# Reproductive Health Services: Attitudes and Practice of Japanese Community Pharmacists

**DOI:** 10.3390/healthcare9101336

**Published:** 2021-10-08

**Authors:** Shigeo Yamamura, Tomoko Terajima, Javiera Navarrete, Christine A. Hughes, Nese Yuksel, Theresa J. Schindel, Tatta Sriboonruang, Puree Anantachoti, Chanthawat Patikorn

**Affiliations:** 1Faculty of Pharmaceutical Sciences, Josai International University, Gumyo 1, Togane, Chiba 283-8555, Japan; terajima@jiu.ac.jp; 2Faculty of Pharmacy and Pharmaceutical Sciences, College of Health Sciences, University of Alberta, Edmonton, AB T6G 2H1, Canada; javiera@ualberta.ca (J.N.); cah1@ualberta.ca (C.A.H.); yuksel@ualberta.ca (N.Y.); terri.schindel@ualberta.ca (T.J.S.); 3Faculty of Pharmaceutical Sciences, Chulalongkorn University, Bangkok 10330, Thailand; tatta.s@pharm.chula.ac.th (T.S.); puree.a@pharm.chula.ac.th (P.A.); Chanthawat.p@gmail.com (C.P.)

**Keywords:** reproductive health, contraception, emergency contraceptives, patient education, community pharmacists

## Abstract

The provision of sexual and reproductive health (SRH) services is an important part of a community pharmacist’s role in many countries. However, such services are not traditionally provided by pharmacists in Japan. We surveyed the practice and attitudes regarding the provision of SRH services among Japanese community pharmacists with a focus on reproductive health (RH) topics. The participants were asked about the provision of RH services, attitudes toward their role as SRH providers, and self-reported confidence in providing education to patients on RH topics. We obtained 534 effective responses. About half of the participants reported providing RH services, and only 21% were involved in dispensing emergency contraception pills. Although the proportion of pharmacists providing education on these topics was considerably lower, about 80% recognized the importance of their role as SRH advisors. Confidence in providing patient education about RH topics depended on their experience in providing such services. Most participants were interested in additional SRH training (80%). Our results suggest that training programs could help to expand Japanese community pharmacists’ roles as SRH providers and increase their confidence in the education of patients. This study provides useful insights to expand pharmacists’ roles in Japan as providers of comprehensive SRH services.

## 1. Introduction

The provision of sexual and reproductive health (SRH) services is an important part of public health and influences the development of a country; it is also a human rights concern [1]. SRH is defined as “a state of physical, emotional, mental, and social well-being in relation to all aspects of sexuality and reproduction” [2]. The World Health Organization (WHO) emphasizes that SRH is part of the global health goals, constituting a special team for promoting research in the SRH field [3]. The WHO vision is the attainment of the highest possible level of SRH for people worldwide.

In a statement issued in 2019, the International Pharmaceutical Federation (FIP) has indicated that pharmacists have the necessary perspective and interest in dealing with gender-related ethical or reproductive health issues [4]. Community pharmacists play an important role in the provision of SRH services, as they are the most accessible healthcare providers in the community. Some specific SRH services, such as chlamydia screening or hormonal contraception-dispensing are provided by community pharmacists in many countries; however, there are many other areas where pharmacists can expand their professional role [5].

One of the essential SRH services offered by community pharmacists is the provision of reproductive health services, such as pregnancy tests, contraception, including emergency contraception (EC), and strategies to prevent sexually transmitted infections. EC reduces the risk of pregnancy after unprotected sexual intercourse or missed or incorrect use of contraceptives [6,7]. In a number of countries, EC pills containing levonorgestrel EC (LNG-EC) are available without a prescription in pharmacies as over-the-counter (OTC) medication or as pharmacist-only access medicines [8,9,10,11].

Healthcare services offered in community pharmacies in Japan are undergoing reform to meet the needs of society [12]. However, in 2018, the ministry of health, labour and welfare decided that EC pills should not change to the OTC category because if a pharmacist sells the product, they need to have specialized knowledge about female reproduction, contraception, and emergency [13]. Japanese community pharmacists have also reported insufficient pharmacy training and a lack of appropriate knowledge and skills to provide enhanced pharmacy services, and therefore, they may feel unprepared to function as providers of SRH services [14]. To expand the role of pharmacists in SRH, it is necessary to assess the attitudes and practice of providing reproductive health services, including EC. In this study, we surveyed Japanese community pharmacists to answer the study question: what are community pharmacists’ practices related to the provision of reproductive health services, attitudes toward these services, and self-reported confidence in providing reproductive health services (including EC) to patients?

This study is part of a broader study aimed at comparing community pharmacists’ perspectives and attitudes regarding the provision of SRH services in Canada, Japan, and Thailand. The purpose of the broader study is to explore and compare the roles and attitudes toward the provision of SRH services by pharmacists in regions with different regulations around pharmacy practices. This report focuses on services related to reproductive health, including pregnancy tests, ovulation tests, contraception, and EC in the Japanese community pharmacy practice context.

### Study Context

In Japan, the barrier contraceptive method, primarily condoms, is the most commonly used to reduce the probability of pregnancy, while the use of oral contraceptive pills is very low (3.0% in 2014) [15].

There are two business models for community pharmacies in Japan: pharmacies (chain or independent) and drugstores. Pharmacists working in settings that represent both models can dispense medicines with prescription. Pharmacies are usually located near clinics or hospitals and focus more on dispensing and compounding. On the other hand, drugstores can also dispense drugs, and focus on selling OTC medicines and miscellaneous products, such as cosmetics and food products.

In Japan, the 6-year Bachelor of Pharmacy program to educate pharmacists was initiated in 2006. The first cohort of graduates were certified as pharmacists based on passing the national examination of pharmacists conducted in 2012 [16,17]. Thus, about 10 years have passed since the introduction of the 6-year pharmacy education program. While the past 4-year Bachelor of Pharmacy program was focused on basic science, the 6-year program is more clinical [18]. Students in the 6-year program receive practical training at both hospital pharmacies and community pharmacies for 11 weeks each [19].

## 2. Materials and Methods

### 2.1. Study Design

This was a descriptive, cross-sectional, observational study. We used a web-based survey to answer the research question. A voluntary, anonymous online survey was distributed via Research Electronic Data Capture (REDCap) [20,21]. An information letter and a consent form were included at the beginning of the survey (Appendix A), completion of the survey and submission of responses implied participant consent. The survey was conducted between November 2020 and April 2021.

### 2.2. Participants

There was no sample size set a priori, a convenience sample of voluntary participants was recruited through email, list distribution, contact lists from pharmacy professional organizations (Ueda, Odawara, Japanese Association for Community Pharmacy), pharmacy chains (Pharcos, Medical system network group), drugstore chains (Welcia Yakkyoku, Aeon), and community pharmacists’ group (Kyoto University SPH, Health informatics pharmacy group). The participants of continuing education programs conducted by the AEON Hapycom Comprehensive Training Organization [22] were also recruited. We also used Twitter and Facebook to recruit pharmacists. Any licensed Japanese pharmacist working in a community setting was able to participate. An initial screening question was used to capture eligible participants.

### 2.3. Data Collection Tool

The survey questions were first developed based on a literature review in English for an international audience. After, it was translated into Japanese. The translated version was refined further to ensure it represented the Japanese context and scope of practice. For face validity testing, the survey was reviewed by experts (*n* = 2, academic pharmacists) and pharmacists (*n* = 5), and then piloted with Canadian pharmacists working in community settings (*n* = 10) and Japanese pharmacists (*n* = 5) [23].

The survey focused on SRH and covered the following topics: pregnancy tests, ovulation tests, contraception (non-hormonal and hormonal), EC, sexually and blood-borne transmitted infections (STBBI), maternal and perinatal health, and general sexual health. However, the focus of this manuscript will be on results regarding reproductive health topics. The questionnaire included pharmacists’ demographic information, educational background, practice regarding reproductive health, attitudes towards providing SRH services, and self-reported confidence in providing education to patients about reproductive health topics (Appendix B).

The primary outcomes were the proportion of pharmacists providing reproductive health, the proportion of pharmacists agreeing (or not) with a series of statements regarding the provision of SRH services, and self-reported confidence in providing reproductive health education to patients. Five-point Likert scales were used to explore attitudes towards providing SRH services and self-reported confidence in providing education on such topics. Additionally, we also assessed the differences in practice between pharmacists working in pharmacies versus drugstores, the influence of pharmacy education on attitudes towards the provision of SRH services, and the relationship between provision of patient education and self-reported confidence.

This study was approved by the research ethics review committee of Josai International (10M200001).

### 2.4. Statistical Analysis

The data were analyzed on JMP-pro version 15 (SAS Institute, Tokyo, Japan). Chi-square test and Fisher’s exact test were used to analyze the association between categorical variables. The Cochran–Armitage test for trend or exact Cochran–Armitage test for trend were used to analyze the trends pertaining to categorical variables. The *p*-value for statistical significance was set at 0.05.

## 3. Results

### 3.1. Participant Characteristics

A total of 743 pharmacists attempted the survey. Of these, 534 (71.9%) were included in the final analysis. A total of 209 (28.1%) possible participants were excluded, 206 (27.7%) because they did not consent and 3 (0.4%) because they did not answer more than 80% of the questionnaire.

### 3.2. Reproductive Health Services Provided by Pharmacists

Most participants were women (55%), younger than 40 years in age (61%), and over half of the participants (51%) had less than 10 years of experience as practicing pharmacists. The majority of participants were employed at corporate chain pharmacies (60%), followed by at drugstores (25%), independent pharmacies (13%), and other types of pharmacies (2%) (Table 1).

Table 2 summarizes the reproductive health services provided by Japanese pharmacists working in different settings. Most pharmacists working in drugstores reported selling pregnancy and ovulation tests (96% and 94%, respectively), but about one-third did not provide patient education on these topics (35% and 26%, respectively). Most participants reported working in a pharmacy which sold barrier contraceptives (95%), however most did not provide patient education on barrier contraceptives (87%). The availability of pregnancy tests and barrier contraceptives reported by pharmacists that worked in drugstores was higher than those working in pharmacy chains and independent pharmacies (96% vs. 42 and 38%, respectively). Almost half of the total samples of pharmacists filled prescriptions for combined hormonal contraceptives (47%). Approximately 20% of all community pharmacists participating in this study dispensed EC medication based on a prescription.

### 3.3. Pharmacists’ Attitudes towards the Provision of SRH Services

About 80% of the participants strongly agreed or agreed with the statement “it is an important part of a community pharmacist’s role to offer advice on sexual and reproductive health”. More than 60% of the participants agreed with the following statements: “community pharmacists should be more involved in sexually transmitted infection prevention, screening, testing, and treatment.”, “as a pharmacist, I have an ethical responsibility to provide SRH services”, and “there is a need to expand the provision of SRH services in the community pharmacy where I work”. The majority of the participants disagreed with the items regarding moral objections (63%), regular use by the community of SRH services offered (58%), and being adequately trained (57%) (Figure 1).

Table 3 summarizes the difference in attitudes towards the provision of SRH service based on the type of bachelor education program. Participants who graduated from the 6-year program tended to agree more with the expansion of the pharmacists’ role in SRH (69% vs. 55%), but also agreed to feeling more embarrassed in giving SRH advice to people (34% vs. 21%). The influence of gender was also analyzed, but no statistical difference was found (data not shown).

### 3.4. Pharmacists’ Self-Reported Confidence to Provide Education on Reproductive Health Topics to Patients

Figure 2 shows participants’ confidence levels in providing education on various reproductive health topics. The reproductive health topics that participants reported more than moderate confidence were as follows: barrier contraception (27.7%), pregnancy tests (27.6%), ovulation tests (24.0%), hormonal contraceptives (22.3%), and EC (16.0%). However, more than half of the participants were only slightly confident or not at all confident when providing education on EC.

Table 4 shows the relationship between the provision of patient education on reproductive health topics and self-reported confidence. Participants who provided reproductive health education services tended to report higher self-confidence scores in educating patients about reproductive health topics (*p* < 0.001 for pregnancy tests, ovulation tests, hormonal contraception, and EC, and *p* = 0.013 for barrier contraceptives for men).

### 3.5. Pharmacists’ Interest in Expanding Their SRH Role and Additional Training

Regarding pharmacists’ role in SRH, two hundred and ninety participants (55%) reported that they would like to expand their role, while 13% indicated not being interested, and 33% did not know. In terms of SRH training, the majority of participants (80%) expressed interest in additional opportunities, and 6% of them would not like to have additional training. There were no statistically significant differences among participants based on educational background differences (data not shown).

## 4. Discussion

This study using a web-based survey revealed current practice and the willingness of Japanese community pharmacists to offer reproductive health services. The results also showed that their confidence to provide education to patients on reproductive health topics is not high, especially regarding EC. The pharmacists who participated in this survey reported that they were interested in expanding their role in SRH services. To achieve that, they need additional training in this regard. This is the first survey on the provision of SRH services and attitudes toward SRH services from Japanese community pharmacists’ perspectives.

Most pharmacists working in drugstores currently sell pregnancy and ovulation test kits and barrier contraceptives, but only one-third of pharmacists working in pharmacies sell these items. This indicates that drugstores, as compared to pharmacies, are more likely to be locations where patients can access these test kits. This is because Japanese pharmacies are more focused on dispensing drugs based on prescriptions. Even among pharmacists working in drugstores, one third sell the test kits without patient education, and only 12% of pharmacists provide patient education for barrier contraceptives. This indicates the inadequacy of the reproductive health services provided by community pharmacists, regarding ensuring safety and proper use of contraceptives. This also implies that community pharmacists need to educate the patients on contraceptive methods because barrier contraception using condoms is the primary method of contraception in Japan [15].

Half of the participants provided the service of dispensing hormonal contraceptives with prescription, and about 20% dispensed EC pills. As a medical doctor’s prescription is required to obtain EC pills in Japan, these pills are more commonly dispensed at clinics or hospitals than at pharmacies. Japanese women are reported to be uncomfortable with obtaining EC pills from pharmacists, or they may not be familiar with the fact that pharmacists dispense EC prescriptions because doctors have historically been permitted to both prescribe and dispense these pills to their patients [24,25]. Therefore, while pharmacists are willing to offer reproductive health services, it can involve a new process that requires adaptation. For example, they probably do not have enough experience dispensing EC pills (as it is commonly prescribed and dispensed by physicians) because they have not been involved with EC dispensing in the past due to national regulations and legislation. In line with these results, participants also reported the need for more training in EC medication. Our survey indicated that training and experience with EC would be expected to increase their confidence to provide reproductive health education to patients [26].

Most participants were willing to offer and expand their roles to provide SRH services, suggesting that Japanese community pharmacists may be attitudinally attuned to providing such services. As 160,000–180,000 abortions are reported per year in Japan [27], it is necessary to provide EC options and sex education for the younger generation. As community pharmacies constitute the most accessible healthcare facilities, the provision of accurate information on reproductive health to the community through such pharmacies would be desirable. Because participants of this study also reported a lack of confidence in providing such services, education and training related to SRH would enable them to expand their role in this field and provide the required services in their community.

Participants who had graduated from the 6-year program were more inclined to provide reproductive health services than those who graduated from the 4-year program. As 10 years have passed since the 6-year program was initiated, most participants who had graduated from the 6-year program were 34 years old or younger when the survey was conducted. Therefore, it is not clear whether this difference that was found was due to the type of pharmacy program undertaken and its curriculum, or the younger age of the participants who had graduated from the 6-year program. Sex education received in primary and secondary schools differs with age; therefore, the age of participants could affect the inclination to provide SRH services [28]. It is also important to consider that sex education provided in Japan may not be adequate and needs improvement [29]. Therefore, pharmacy education could be a stronger influence on whether the participants were willing to provide SRH services or not.

More than 80% of the participants who provided SRH services showed at least some confidence in providing education to patients about reproductive health topics. This suggested that prior experience raises their confidence in delivering such services. Most participants were interested in additional SRH training. Still, only 55% reported a desire to expand their role in SRH services, suggesting that they may not have enough confidence to provide SRH services or they may need additional support, for example, offered by pharmacy owners (such as installation of consulting rooms) or pharmacist organizations (additional training or education opportunities).

This study had some limitations. The generalizability of the results is limited because of voluntary response bias and non-response bias. The number of participants in the younger age group was higher than that of pharmacists in Japan. As the participants would have more interest in providing SRH services, this would imply a bias towards including more pharmacists who wanted to expand their role in SRH services.

Different strategies were used to distribute the survey, and participants were not identified in any way, so this makes it challenging to identify if pharmacists completed the questionnaire more than once. Furthermore, using Facebook and Twitter as recruitment platforms could lead to selection biases towards younger pharmacists. However, this could be mitigated by using other recruitment venues (pharmacists associations, pharmacy chains, and drugstores). 

The characteristics of the Japanese cohort indicate that the survey represented a particular group of pharmacists, and the results may not be generalizable to all Japanese pharmacists. The sample of this study may not be representative of Japanese pharmacists because the proportion of women was slightly lower (55% vs. 61%), and the proportion of pharmacists aged <40 years old was higher (61% vs. 38%) [30]. However, it is relevant to mention that this survey aimed to target community pharmacists only, and there are no available national demographic statistics for the subgroup of Japanese pharmacists. Additionally, there was no sample size set a priori, and we approached this by using convenience sampling.

Despite these limitations, there are some strengths to this study. This is the first survey addressing the current situation and practice of reproductive health services from Japanese community pharmacists’ perspectives. This study is also the first to reveal that most Japanese community pharmacists have a positive attitude toward providing SRH services and are interested in expanding their role and having additional training in SRH.

Differences in practices, attitudes towards sex or sexual health, and confidence on these topics might be partially influenced by experience and education as well as personal beliefs. It is necessary to expand pharmacists’ roles beyond providing traditional product-focused services in several SRH areas. Further research could look into strategies to support the expansion of pharmacists’ roles and the incorporation of comprehensive SRH pharmacy services into practice, as well as the impact of the SRH services that pharmacists provide to their communities.

## 5. Conclusions

To our knowledge, this is the first Japanese study to include several SRH topics and address them from a pharmacists’ perspective lens. The findings of this study indicate that pharmacists are involved in the delivery of SRH pharmacy services to varying extents. While Japanese community pharmacists are willing to offer reproductive health services, they do not feel confident enough to provide patient education on reproductive health topics, especially EC. Japanese community pharmacists are interested in expanding their role and receiving additional training on reproductive health topics. Education, experience, and training seem necessary to achieve that goal. The results of this survey can guide future studies to explore the reasons for disengagement of pharmacists in reproductive health services, and ways to support the incorporation of comprehensive SRH pharmacy services into practice.

## Figures and Tables

**Figure 1 healthcare-09-01336-f001:**
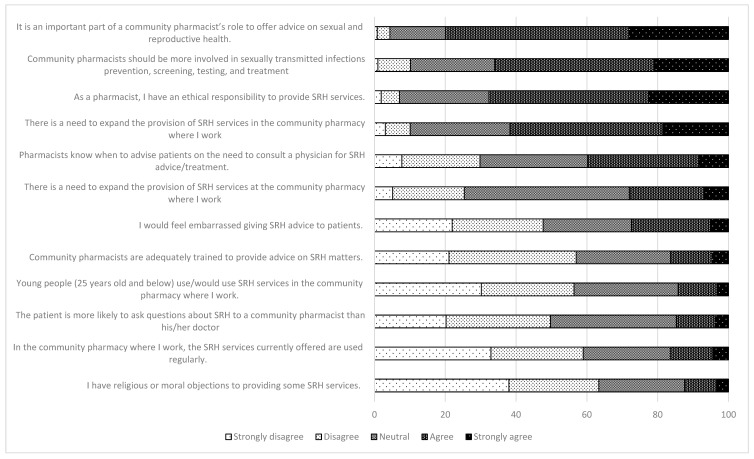
Attitudes toward the provision of sexual and reproductive health (SRH) services. SRH: sexual and reproductive health.

**Figure 2 healthcare-09-01336-f002:**
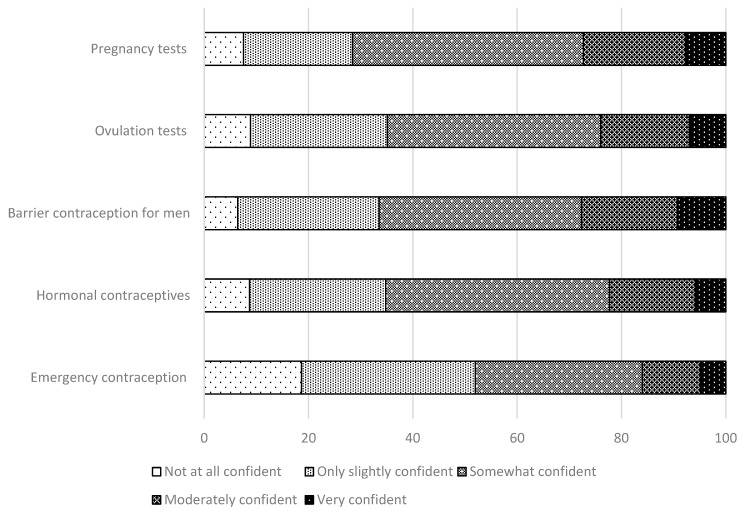
Confidence to provide education on topics related to reproductive health.

**Table 1 healthcare-09-01336-t001:** Background characteristics of participants.

Characteristics	*n* (%)
Gender	
Female	295 (55)
Age range	
20–30 years	208 (39)
31–40 years	116 (22)
41–50 years	97 (18)
51–60 years	81 (15)
61–70 years	29 (5)
71+ years	3 (1)
Professional Education	
Bachelor of Pharmacy (4 years)	245 (46)
Bachelor of Pharmacy (6 years)	257 (48)
Master (MSc or MPharm)	23 (4)
Doctor of Philosophy (PhD)	6 (1)
Years registered as a pharmacist	
<1 year	9 (2)
1–5 years	214 (40)
6–10 years	50 (9)
11–20 years	91 (17)
21–30 years	95 (18)
>31 years	74 (14)
Type of pharmacy	
Independent	68 (13)
Corporate/chain	323 (60)
Drugstore chain	134 (25)
Others	9 (2)

**Table 2 healthcare-09-01336-t002:** Reproductive health services provided by Japanese community pharmacists.

Service	Total (*n* = 534) (%)	Drugstore (*n* = 134) (%)	Corporate/Chain (*n* = 323) (%)	Independent pharmacy (*n* = 68) (%)	Others (*n* = 9) (%)
Pregnancy Tests					
Does your pharmacy sell pregnancy tests?	293 (55)	127 (96)	134 (42)	26 (38)	6 (75)
Do you provide patient education on pregnancy tests?	192 (36)	87 (65)	90 (28)	9 (13)	6 (67)
Ovulation Tests					
Does your pharmacy sell ovulation tests?	297 (56)	127 (94)	147 (46)	18 (26)	5 (63)
Do you provide patient education on ovulation tests?	222 (42)	98 (74)	107 (33)	13 (25)	4 (44)
Contraception					
Does your pharmacy sell male non-hormonal (barrier) contraceptives? (e.g., condoms)	244 (46)	126 (95)	97 (31)	17 (26)	4 (57)
Do you provide patient education on male non-hormonal (barrier) contraceptives?	37 (7)	17 (13)	14 (4)	3 (4)	3 (33)
Do you dispense combined hormonal contraceptives? (with prescription)	250 (47)	80 (60)	141 (44)	24 (35)	5 (56)
Do you provide patient education on hormonal contraception?	207 (39)	77 (57)	109 (34)	16 (24)	5 (56)
Emergency contraception					
Do you dispense emergency contraception pills (ECPs)?	111 (21)	19 (16)	85 (27)	5 (7)	2 (25)
Do you provide patient education on ECPs?	81 (15)	20 (15)	51 (16)	6 (9)	4 (44)

Numbers of pharmacists who answered “yes” are presented, and the percentage is included in parentheses.

**Table 3 healthcare-09-01336-t003:** Statistically significant items regarding attitudes towards the provision of sexual and reproductive health (SRH) services (detailed in Figure 1), based on education programs.

**There is a need to expand the provision of sexual and reproductive health services in this pharmacy (*p* = 0.002) ***						
	**Strongly disagree**	**Disagree**	**Neutral**	**Agree**	**Strongly agree**	**Total**
Bachelor of Pharmacy (4 years)	9 (4)	25 (10)	74 (31)	100 (42)	31 (13)	239
Bachelor of Pharmacy (6 years)	6 (2)	10 (4)	64 (25)	116 (46)	58 (23)	254
**There is a need for sexual health and reproductive services in the local area near this pharmacy (*p* = 0.031)**						
	**Strongly disagree**	**Disagree**	**Neutral**	**Agree**	**Strongly agree**	**Total**
Bachelor of Pharmacy (4 years)	18 (7)	113 (46)	70 (29)	37 (15)	6 (2)	244
Bachelor of Pharmacy (6 years)	6 (2)	118 (47)	34 (13)	69 (27)	25 (10)	252
**I would be embarrassed giving sexual and reproductive health advice to people (*p* = 0.007) ***						
	**Strongly disagree**	**Disagree**	**Neutral**	**Agree**	**Strongly agree**	**Total**
Bachelor of Pharmacy (4 years)	59 (24)	58 (24)	73 (30)	44 (18)	8 (3)	242
Bachelor of Pharmacy (6 years)	51 (20)	66 (26)	51 (20)	69 (27)	18 (7)	255
**Community pharmacists are adequately trained to provide advice on sexual and reproductive health matters (*p* < 0.001)**						
	**Strongly disagree**	**Disagree**	**Neutral**	**Agree**	**Strongly agree**	**Total**
Bachelor of Pharmacy (4 years)	76 (31)	85 (35)	67 (27)	12 (5)	5 (2)	245
Bachelor of Pharmacy (6 years)	30 (12)	98 (38)	66 (26)	42 (16)	19 (7)	255
**Young people (aged 25 years and below) would use sexual and reproductive health services at this pharmacy (*p* < 0.001)**						
	**Strongly disagree**	**Disagree**	**Neutral**	**Agree**	**Strongly agree**	**Total**
Bachelor of Pharmacy (4 years)	114 (48)	10 (4)	57 (24)	58 (24)	1 (0)	240
Bachelor of Pharmacy (6 years)	39 (15)	44 (17)	88 (34)	70 (27)	15 (6)	256

Numbers and percentages in parentheses. *p*-values obtained using Fisher’s exact test, except for * in the chi-square test.

**Table 4 healthcare-09-01336-t004:** Relationship between the provision of patient education with reference to sexual and reproductive health (SRH) services, and self-reported confidence.

Patient Education Service						
Pregnancy tests (*p*-value: chi-square test: <0.001, Cochran–Armitage test for trend: <0.001)						
Patient education	Confident					
	Very	Moderately	Somewhat	Only slightly	Not at all	Total
Provided	25	45	92	25	3	190
Not provided	16	59	143	86	37	341
Ovulation tests (*p*-value: chi-square test: <0.001, Cochran–Armitage test for trend: <0.001)						
	Very	Moderately	Somewhat	Only slightly	Not at all	Total
Provided	22	56	104	32	6	220
Not provided	14	33	110	107	41	305
Barrier contraceptives for men (*p*-value: Fisher’s exact e test: 0.0323, exact Cochran–Armitage test for trend: 0.013)						
	Very	Moderately	Somewhat	Only slightly	Not at all	Total
Provided	5	9	18	3	1	36
Not provided	44	88	185	140	33	490
Hormonal contraception (*p*-value: chi-square test: <0.001, Cochran–Armitage test for trend; <0.001)						
	Very	Moderately	Somewhat	Only slightly	Not at all	Total
Provided	18	45	96	40	6	205
Not provided	13	42	131	98	40	324
Emergency contraception (*p*-value: chi-square test: <0.001, Cochran–Armitage test for trend: <0.001)						
	Very	Moderately	Somewhat	Only slightly	Not at all	Total
Provided	7	19	30	18	7	81
Not provided	19	39	136	158	92	444

## Data Availability

The data presented in this study are available on request from the corresponding author.

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
