# Peer review of "Reproductive Health Services: Attitudes and Practice of Japanese Community Pharmacists"

_healthcare, 2021, doi:10.3390/healthcare9101336_

Round 1

Reviewer 1 Report

The article entitled “Reproductive Health Services: Attitudes and Practice of Japanese Community Pharmacists” addresses a socially important topic regarding the provision of SRH services among Japanese practising pharmacists

I have a few minor comments:

  1. The authors write (p. 2, lines 60-61): „This may be partly because it is believed that Japanese community pharmacists are unprepared to function as providers of SRH services "- who thinks so? but this is due not to the right-wing government's pro-family policy based on beliefs and religion that prevails in a given country. Many reproductive health professionals protest against such ordinances, and there is a group that supports such decisions, of course. However, there is no such thing as related to beliefs about pharmacists' knowledge (which, of course, may not be sufficient.) Therefore, I recommend the authors clarify whether this is their personal beliefs or the assumption that pharmacists are insufficiently prepared for such counselling results, for example, from some social discussions in Japan or other research.
  2. In the Materials and Methods section, the authors write that they used direct mailing to pharmacists meeting certain criteria, as well as social network media for recruitment. However, there is no detailed information on the study procedure. Were the people who expressed their willingness to participate in the study contacted the researchers in order to receive a personalized link, or was the link to the study immediately included in the information about the study and could be filled in by anyone who believed that they met the research criteria? This is important information because it is not known from the current description whether the researchers controlled in some way, e.g. whether none of the participants completed the questionnaire more than once, etc.
  3. How did the authors determine sample size and effect size for the Japanese group?
  4. The authors report that "The survey was subjected to validity testing" (p. 3, line 109) - what are the results of this validation for the second part of the survey entitled "Attitudes toward Sexual and Reproductive Health (SRH) services". Please provide at least Cronbach's alpha for the current study sample.
  5. To make the content of the methodological part more readable, I recommend introducing sub-titles for e.g. Statistical analysis, Instruments, etc. rather than just starting the next sections with a new paragraph.
  6. In each case where the authors provide the information "data not shown" in brackets, I recommend that you provide the data in supplementary materials or open data repository (and provide access to this data in the article).

Author Response

Thank you for reviewing our manuscript and thoughtful comments and suggestions. Please see attached file for our responses to your comments.

Reviewer 2 Report

Thank you for the submission. The manuscript is very long for such a simple study. The introduction could be much shorter. Tables 3, 4 and 5 are unnecessary; the significant information could be summarised in the text. The whole study (including all countries) could very easily be within one manuscript. It certainly does not warrant 4 publications (one for each country and a comparison of the 3 countries), if that is the plan.

The abstract is unclear (one sentence says half of participants provided SRH while the next sentence refers to 80%): “Half of the participants provided SRH services but did not provide patient education in all cases; 20% were involved in dispensing emergency contraception pills. About 80% of the participants strongly agreed or agreed to provide SRH services and were involved in SRH.”

The recruitment is unclear : “..and those who worked at selected community pharmacy chains and drugstore chains.”  What does ‘selected’ mean?

The response rate was not 71.9%. How many pharmacists were invited to participate? Were the respondents representative of all Japanese pharmacists? Based on gender, it seems not. Later, it is acknowledged that the sample was not representative, casting some doubt on the results.

Author Response

Thank for your reviewing and thoughtful comments and suggestions. Please see the attached file for our responses to your comments. 

Round 2

Reviewer 2 Report

Thank you for the revision.